# Associations of the *SREBF2* Gene and *INSIG2* Polymorphisms with Obesity and Dyslipidemia in Thai Psychotic Disorder Patients Treated with Risperidone

**DOI:** 10.3390/jpm11100943

**Published:** 2021-09-22

**Authors:** Natchaya Vanwong, Chonlaphat Sukasem, Weerapon Unaharassamee, Napa Jiratjintana, Chalitpon Na Nakorn, Yaowaluck Hongkaew, Apichaya Puangpetch

**Affiliations:** 1Department of Clinical Chemistry, Faculty of Allied Health Sciences, Chulalongkorn University, Bangkok 10330, Thailand; natchaya.v@chula.ac.th; 2Cardiovascular Precision Medicine Research Group, Department of Clinical Chemistry, Faculty of Allied Health Sciences, Chulalongkorn University, Bangkok 10330, Thailand; 3Division of Pharmacogenomics and Personalized Medicine, Department of Pathology, Faculty of Medicine Ramathibodi Hospital, Mahidol University, Bangkok 10400, Thailand; chonlaphat.suk@mahidol.ac.th; 4Laboratory for Pharmacogenomics, Somdech Phra Debaratana Medical Center (SDMC), Ramathibodi Hospital, Bangkok 10400, Thailand; 5Pharmacogenomics and Precision Medicine, The Preventive Genomics & Family Check-Up Services Center, Bumrungrad International Hospital, Bangkok 10110, Thailand; 6Department of Psychiatry, Somdet Chaopraya Institute of Psychiatry, Bangkok 10600, Thailand; UWEERAPON@gmail.com (W.U.); noknapaj@yahoo.com (N.J.); 7Department of Clinical Pharmacy, Faculty of Pharmaceutical Sciences, Prince of Songkla University, Songkhla 90110, Thailand; nchalitpon@gmail.com; 8Advance Research and Development Laboratory, Bumrungrad International Hospital, Bangkok 10110, Thailand; yaowaluck.hong@outlook.com

**Keywords:** *SREBF2* gene polymorphism, *INSIG2* polymorphism, risperidone, obesity, dyslipidemia

## Abstract

Background: Patients with psychotic disorders who receive atypical antipsychotic drugs often develop metabolic abnormalities. The sterol regulatory element-binding factor 2 (SREBF2) gene and insulin-induced gene (INSIG) have important roles in lipid metabolism. A previous study indicated that risperidone stimulated both lipogenesis and cholesterogenesis through activation of SREBP2 expression and inhibition of INSIG2. The *SREBF2* gene and *INSIG2* polymorphisms have been reported to be associated with metabolic abnormalities. Objective: To investigate the association of the *SREBF2* gene (rs1052717, rs2267439, and rs2267443) and *INSIG2* (rs7566605, rs11123469, and rs17587100) polymorphisms and the presence of obesity and dyslipidemia in Thai psychotic disorder patients treated with risperidone. Methods: All 113 psychiatric patients using risperidone were evaluated for their lipid profile and screened for obesity criteria. We genotyped the *SREBF2* gene and *INSIG2* polymorphisms using TaqMan real-time polymerase chain reaction. Results: None of the studied *SREBF2* gene and *INSIG2* SNPs were associated with obesity in Thai psychotic disorder patients receiving risperidone. Nonetheless, the *SREBF2* rs2267443 (G/A) A-allele carriers were at a higher risk for hypertriglyceridemia, whereas the *INSIG2* rs11123469 (T/C) C-allele carriers had a lower risk for hypertriglyceridemia, after being adjusted for clinical characteristics using multiple logistic regression. Conclusions: Our findings suggest that the *SREBF2* gene rs2267443 (G/A) and the *INSIG2* rs11123469 (T/C) polymorphisms are associated with dyslipidemia in Thai psychotic disorder patients treated with risperidone. Further studies with prospective designs and larger patient groups are needed.

## 1. Introduction

Risperidone is a commonly prescribed atypical antipsychotic (AAP) drug used for the treatment of schizophrenia and other psychotic disorders. Recently, there has been concern regarding many atypical antipsychotic drugs because of their propensity to induce metabolic disturbances such as type 2 diabetes, weight gain, dyslipidemia, and metabolic syndrome [1,2,3], which increase the risk of cardiovascular diseases [4,5]. Moreover, patients with psychotic disorders have two times the risk of cardiovascular disease (CVD) mortality as compared with people in the general population [6,7]. Metabolic adverse effects are risk factors for cardiovascular disease that lead to increased morbidity and mortality and cause nonadherence with AAP medication [8], which leads to a poor health-related quality of life for people with psychotic disorders [9]. The mechanisms underlying AAP-induced metabolic adverse effects remain poorly understood. The high interindividual differences of the occurrence of metabolic effects in psychotic disorders may be due to genetic predisposition, medication effects, or lifestyle. Since genetic factors have been proposed to be involved [10,11], a better understanding of the genetic risk factors involved in the metabolic pathways may help pretreatment selection and lead to more individualized treatments based on their genetic profiles. Previously, many genes have been proposed to be related to metabolic dysfunction, and the influences of genetics may be different across racial and ethnic groups [10,11]. However, few studies have observed the impacts of genes involved in the SREBPs in lipid metabolism and the insulin-signaling (SREBP/INSIG) pathway [12,13] in patients who received AAP drugs, and therefore, the results are still inconclusive regarding the association of SREBP/INSIG pathway genes with metabolic disturbances in psychotic disorders treated with AAP drugs.

Sterol regulatory element-binding transcription proteins (SREBPs) are a family of transcription factors that regulate lipid biosynthesis, adipogenesis, and lipid homoeostasis [14,15]. SREBPs form complexes with SREBP cleavage-activating protein (SCAP), a transport protein that binds to sterol regulatory element-binding proteins (SREBPs) and mediates their transport from the endoplasmic reticulum (ER) to the Golgi apparatus, where inactivated SREBPs are activated before entering nuclear and regulating genes for lipid biosynthesis. For regulation of the lipid metabolism, when the cholesterol levels are high, insulin-induced genes (INSIGs) are bound to SCAP to promote the ER retention of SCAP and block SCAP/SREBP transportation from the ER to the Golgi and, consequently, to decrease cholesterogenesis and lipogenesis [16]. A previous study indicated that risperidone and clozapine stimulated both lipogenesis and cholesterogenesis through the activation of SREBP2 expression and inhibition of INSIG2 [17]. Since obesity always co-occurs with dyslipidemia, the changes in the metabolism of adipocyte, owing to the polymorphisms in the *SREBP2* gene and/or *INSIG2*, may lead to obesity and/or dyslipidemia. The *SREBF2* gene and *INSIG2* may be candidate genes for risperidone-induced dyslipidemia, obesity, and cardiovascular diseases in psychotic disorder patients. Recently, three single nucleotide polymorphisms (SNPs) of the *SREBP2* gene in the intronic region, i.e., IVS11 + 414 G > A (rs1052717), IVS1 + 8408 T > C (rs2267439), and IVS12-1667 G > A (rs2267443), have been associated with T2DM [18] and metabolic syndrome [12]. In addition, *INSIG2* presents three SNPs in the intronic region, i.e., rs7566605 (G/C), rs11123469 (T/C), and rs17587100 (A/C), which have been associated with an increased risk for weight gain [19,20] and metabolic syndrome [13]. Considering the prominent role of these genes in metabolic adverse effects, we hypothesized that the genetic *SREBF2* gene and *INSIG2* polymorphisms might be involved in the increased risk of obesity and dyslipidemia in Thai patients treated with risperidone. The objective of this study was to examine the association of the *SREBF2* gene (rs1052717, rs2267439, and rs2267443) and *INSIG2* (rs7566605, rs11123469, and rs17587100) polymorphisms on the presence of obesity and dyslipidemia in Thai psychotic disorder patients treated with risperidone.

## 2. Materials and Methods

### 2.1. Eligible Patients

A total of 113 psychotic disorder patients were recruited from the Somdet Chaopraya Institute of Psychiatry in Central Thailand. All psychotic disorder patients were diagnosed by a psychiatrist. All subjects had received risperidone for at least 3 months. Patients who took other psychotropic drugs, lithium, or sodium valproate were excluded. All participants were screened for the following: age, gender, types of psychotic disorders, duration of treatment, dosage regimen, and smoking status. This study was approved by the Ethics Committee of the Faculty of Medicine, Ramathibodi Hospital, Mahidol University, Bangkok, Thailand. The study protocol was clearly explained to all patients, and informed consent was given before the study.

### 2.2. Biochemical and Anthropometric Assessments

Anthropometric (including weight, height, body mass index (BMI), and waist circumference) and lipid (including total cholesterol levels, triglyceride levels, LDL cholesterol levels, and HDL cholesterol levels) profiles were measured in the current study. The presence of obesity according to the WPRO standard was proposed by the International Association for the Study of Obesity and the IOTF in 2000 [21] in order to have more applicable and appropriate BMI cut-off points for Asian populations. According to the WPRO criteria, a BMI ≥ 25.0 kg/m^2^ is classified as obese. A waist circumference ≥ 90 cm in men or ≥80 cm in women is defined as abdominal obesity. The criteria for dyslipidemia include (1) fasting serum total cholesterol levels ≥ 200 mg/dL, (2) fasting serum triglyceride levels ≥ 150 mg/dL, (3) fasting serum LDL-C levels ≥ 130 mg/dL, and (4) fasting serum HDL-C levels < 40 mg/dL in men or <50 mg/dL in women.

### 2.3. Genomic DNA Preparation and Genotyping Assay

Genomic DNA was extracted from EDTA blood samples using an automated MagNA Pure Compact (Roche Applied Science, Penzberg, Germany). The DNA concentration and purification were measured with a NanoDrop spectrophotometer (Thermo Fisher Scientific, Wilmington, DE, USA). A total of 6 SNPs in the *SREBF2* gene (rs1052717, rs2267439, and rs2267443) and *INSIG2* (rs7566605, rs11123469, and rs17587100) were genotyped using TaqMan^®^ Genotyping Assays (Applied Biosystems™, Carlsbad, CA, USA), according to the manufacturer’s instructions.

### 2.4. Statistical Analysis

Descriptive statistics were used to describe the clinical characteristics of the subjects. Data are reported as the median (interquartile range, IQR). Accordingly, for nonparametric data, the Mann–Whitney *U* test compared the differences in median values (IQR) between the two groups. The chi-square test was used for comparisons between two categorical variables. Since genetic and nongenetic factors, including age, gender, types of psychotic disorders, duration of risperidone treatment, risperidone dose, and smoking status, might be risk factors for dyslipidemia, a multiple logistic regression model was constructed to find the factors independently associated with dyslipidemia. All statistics were calculated using SPSS software version 22 (Chicago, IL, USA), and the statistical significance was set at *p* < 0.05.

## 3. Results

### 3.1. Clinical Characteristics and Genotyping Data

A total of 113 patients with psychotic disorders were genotyped for the *SREBF2* gene and the *INSIG2* polymorphisms. The genotype and allele frequencies of the *SREBF2* gene (rs1052717, rs2267439, and rs2267443) and the *INSIG2* (rs7566605, rs11123469, and rs17587100) polymorphisms are listed in Table 1. The summary information of the distribution of the demographic and clinical characteristics is shown in Table 2. The results were examined according to the absence or presence of obesity. About 50% (57/113) of the patients were in the obese group. Among the impacts of the demographic and clinical characteristics, there were no relationships between obesity and the demographic and clinical characteristics.

### 3.2. Effect of the SREBF2 Gene (Rs1052717, Rs2267439, and Rs2267443) Polymorphisms on Obesity and Dyslipidemia in Patients with Psychotic Disorders

In this study, we investigated the association of the *SREBF2* gene (rs1052717, rs2267439, and rs2267443) polymorphisms with obesity and dyslipidemia; a relationship between dyslipidemia and the *SREBF2* gene rs2267443 (G/A) polymorphism was found. Patients with the *SREBF2* gene rs2267443 (GA + AA) genotype had a significantly higher incidence of hypertriglyceridemia as compared with the GG genotype (odds ratio = 2.56, 95% confidence interval = 1.12–5.99, *p* = 0.02) (Table 3). However, there was no association between the *SREBF2* gene rs2267443 polymorphism and obesity. In addition, there was no association between obesity or dyslipidemia with the *SREBF2* rs1052717 (G/A) or *SREBF2* rs2267439 (T/C) polymorphisms.

### 3.3. Effect of INSIG2 (Rs7566605, Rs11123469, and Rs17587100) Polymorphisms on Obesity and Dyslipidemia in Patients with Psychotic Disorders

The associations between the *INSIG2* (rs7566605, rs11123469, and rs17587100) polymorphisms and obesity and dyslipidemia in patients with psychotic disorders are shown in Table 4. There was a borderline significant association between the *INSIG2* rs11123469 (T/C) polymorphism and hypertriglyceridemia (odds ratio = 0.54, 95% confidence interval = 0.27–1.06, *p* = 0.07). However, there was no association between the *INSIG2* rs11123469 polymorphism and obesity. Additionally, there was no relationship between obesity or dyslipidemia with the *INSIG2* rs7566605 (G/C) or *INSIG2* rs17587100 (A/C) polymorphisms.

### 3.4. Multivariate Logistic Regression Analysis of Predictive Factors for Risperidone-Induced Hypertriglyceridemia in Patients with Psychotic Disorders

A multivariate logistic regression analysis was performed to analyze the association of risperidone-induced hypertriglyceridemia in psychotic disorder patients with genetic factors and nongenetic factors (Table 5). The association of hypertriglyceridemia in patients with the *SREBF2* rs2267443 (G/A) polymorphism remained significant after adjustment by multiple logistic regression (odds ratio = 4.47, 95% confidence interval = 1.62–12.32, *p* = 0.004). Moreover, the multivariate model revealed that the polymorphism of *INSIG2* rs11123469 (T/C) was found to be significantly associated with hypertriglyceridemia after adjustment for the multivariate analyses (odds ratio = 0.37, 95% confidence interval = 0.14–0.97, *p* = 0.043).

## 4. Discussion

AAP drugs have a broader spectrum of efficacy and have a lower propensity to induce EPS and tardive dyskinesia as compared with typical antipsychotic (TAP) drugs [22]. Over the past decade, AAP drugs have been extensively used for the treatment of psychotic disorders in preference to TAP drugs. Nevertheless, AAP drugs cause metabolic adverse effects such as dyslipidemia and metabolic syndrome with an increased risk of consequent cardiovascular disease [23,24]. Patients with psychotic disorders who take AAP drugs are known to be at a high risk of developing metabolic dysfunction, which leads to cardiovascular morbidity and mortality [23,25]. A study by M. Murashita et al. revealed that patients on long-term risperidone monotherapy had significantly impaired body fat percentage, BMI, triglyceride, and HDL cholesterol levels as compared with healthy volunteers [26]. Consistently, a retrospective study found that risperidone-treated patients had significant weight increases, BMI, and fasting triglycerides after 1 year of treatment as compared with a baseline [27]. In addition, a prospective cohort study reported that there was a significant increase in body weight and triglyceride levels in the first month, which continued in the second month, after patients were treated with risperidone [28]. The exact mechanism for risperidone-related metabolic adverse effects is still inconclusive. Identification of the genetic polymorphisms that affect drug-related metabolic adversities may help to clarify the mechanisms underlying the metabolic abnormalities and may assist in the selection of more suitable drugs. To the best of our knowledge, this is the first study to examine the association of the *SREBF2* gene (rs1052717, rs2267439, and rs2267443) and *INSIG2* (rs7566605, rs11123469, and rs17587100) polymorphisms and the presence of obesity and dyslipidemia in Thai psychotic disorder patients treated with risperidone. In our psychotic disorder sample, we found that half of the patients were obese. Since obesity is the consequence of environmental and gene interactions, our study determined the contribution of nongenetic variables to obesity in Thai patients with psychotic disorders. The result reveals that nongenetic factors (including age, gender, types of psychotic disorders, duration of treatment, dosage, and smoking status) are not significantly associated with obesity in Thai psychotic disorder patients receiving risperidone (Table 1). In regard to the roles of SREBP2 and INSIG2 in lipid metabolism regulation, it is more likely that obesity and dyslipidemia susceptibility may be related to the polymorphisms of the *SREBP2* gene and *INSIG2*.

The *SREBP2* gene encodes the membrane-bound transcription factor that controls lipid homeostasis [29]. Previous studies have shown that SREBP expression was associated with a cardiovascular risk factor for the severity of CAD and poor lipid control [30]. In this study, we investigated the association of *SREBF2* gene polymorphisms with obesity and dyslipidemia in psychotic disorder patients treated with risperidone. The results showed that none of the *SREBF2* gene (rs1052717, rs2267439, and rs2267443) polymorphisms were related to the obesity in Thai psychotic disorder patients receiving risperidone. Nevertheless, our results provided evidence of an association between the *SREBF2* rs2267443 (G/A) polymorphism and hypertriglyceridemia. Patients carrying the A allele of the *SREBF2* (rs2267443) gene were at an increased risk for hypertriglyceridemia (after adjustment for the multivariate analysis). Consistent with our findings, Yang et al. reported that patients receiving the AAP drug who carried the *SREBF2* (rs2267443) A allele were at an increased risk for metabolic syndrome after controlling for a potentially confounding effect [12]. In contrast, Galavi et al., who studied T2DM patients, reported that there was an association of the *SREBF2* (rs2267439) polymorphism with decreased HDL cholesterol, while the *SREBF2* (rs1052717 and rs2267443) SNPs were not associated with the plasma lipids levels [18].

Concerning *INSIG2*, since *INSIG2* has been functionally linked to lipid metabolism, owing to its role in endogenous cholesterol and fatty acid synthesis feedback inhibition [16], *INSIG2* might be a candidate gene for obesity and dyslipidemia. A previous study demonstrated that the weights of INSIG2 knockout mice were more than that of the controls, and the INSIG2 knockout mice had a higher accumulation of triglycerides and cholesterol in the livers [31]. In addition, INSIG2 is also expressed in adipocytes, and this expression is enhanced during the regulation of adipocyte [32]. Furthermore, an integrating quantitative trait loci and high-density SNP analyses in mice identified *INSIG2* as a strong susceptibility gene for the plasma cholesterol levels [33]. Many studies have revealed a significant association between obesity or BMI and the *INSIG2* (rs7566605) polymorphism [20,34]. A genome-wide association study demonstrated that a SNP rs7566605 at 10-kb upstream of the transcription start site of INSIG2 was related to adult and childhood obesity [20]. Nevertheless, several other studies have failed to show a relationship between obesity and the *INSIG2* (rs7566605) polymorphism [35,36,37], which is consistent with our study. Since dyslipidemia often occurs alongside obesity, previous studies have found a relationship between the *INSIG2* (rs7566605) polymorphism and lipid markers (total cholesterol, triglycerides low-density lipoproteins (LDL), and/or high-density lipoproteins (HDL)); however, there have been inconsistent findings [36,38]. In our study, we found that there was no association between the *INSIG2* (rs7566605) polymorphism and obesity or dyslipidemia in Thai psychotic disorder patients receiving risperidone. Therefore, we propose that *INSIG2* (rs7566605) may not be a causal variant; other genetic polymorphisms might play an important role in the development of obesity and/or dyslipidemia. Le Hellard and colleagues revealed a strong association between the *INSIG2* (rs17587100) polymorphism and weight gain in patients receiving antipsychotic drugs [19]. In contrast, our study observed a nonsignificant association with *INSIG2* (rs17587100) and risperidone-induced obesity and/or dyslipidemia. Similarly, Tiwari et al. revealed that *INSIG2* (rs17587100) was not associated with an AAP-induced weight gain in African American and European schizophrenia patients [39]. Moreover, in this study, we did not find an association between *INSIG2* rs11123469 and obesity. Likewise, Liou et al. found that there was no association between *INSIG2* rs11123469 and BMI changes in patients treated with AAP drugs [13]. In regard to INSIG2 playing an essential role in the regulation of lipid metabolism, Liou and colleagues demonstrated that the C allele of the *INSIG2* (rs11123469) polymorphism was significantly overrepresented in those with metabolic syndrome in patients treated with AAP drugs [13]. In contrast, in this study, we found that the protective effect of *INSIG2* rs11123469 (T/C) was associated with hypertriglyceridemia. Patients carrying the C allele of *INSIG2* rs11123469 were at a lower risk for hypertriglyceridemia after controlling for potential confounders. Indeed, in our population, individuals with hypertriglyceridemia more often carried the common alleles (T allele) of *INSIG2* rs11123469 (T/C), whereas those without hypertriglyceridemia more often carried the minor alleles (C allele). M. Arca et al. revealed that the elevation of triglyceride was associated with a significantly increased risk of all-cause mortality and atherosclerotic cardiovascular disease (ASCVD) events [40]. Triglyceride lowering may lead to a decreased risk of ASCVD events.

There are various variables that may explain the discrepancies in the results across studies: (1) The studied populations with different antipsychotic drugs used different doses of medication; therefore, the heterogeneity of the medications taken by patients may have been a confounding factor. (2) The ethnic differences in the minor allele frequency of the studied SNPs, as well as other genetic risk factors, may influence obesity and dyslipidemia. Further studies are required to determine the impact of the SREBF2 and INSIG2 allelic forms on the response of obesity and dyslipidemia in the cells of rats exposed to risperidone for the elucidation of the functional consequence of the *SREBF2* gene and *INSIG2* polymorphisms. Our findings suggest that the *SREBF2* gene (rs2267443) and the *INSIG2* (rs11123469) polymorphisms may contribute to the underlying pathophysiology of dyslipidemia in patients receiving risperidone, which supports a previous study that risperidone exposure affected the SREBP-regulated lipid biosynthesis and other lipid homeostasis pathways [17].

This is the first study of the association between the *SREBF2* gene and *INSIG2* polymorphisms and obesity and dyslipidemia in Thai psychotic disorder patients treated with risperidone. Further studies are needed to assess the role of the *SREBF2* gene and *INSIG2* in obesity and dyslipidemia. This study has some limitations. First, this study is a cross-sectional study; the changes in the metabolic parameters over time related to the use of risperidone could not be analyzed. Thus, in order to establish causality, longitudinal studies should be conducted. Second, based on the experimental design, our genetic analysis did not control for lifestyle (e.g., food intake and physical activity) factors because of limited data. Nevertheless, by stratification analysis, there was no effect of age, gender, disease, duration, dosage, or smoking on our significant findings. Third, this study only genotyped a few SNPs that previously showed significant results in a genome-wide association study or association studies. Other genetic risk factors may have an effect on antipsychotic-induced metabolic effects in psychotic disorder patients. Finally, the number of subjects in this study was rather small, which may have limited the power to determine the association between the genetic polymorphisms and obesity and dyslipidemia in patients receiving risperidone; therefore, the results must be replicated in a larger population.

## 5. Conclusions

In conclusion, this is the first study of the relationship between the *SREBF2* gene and *INSIG2* polymorphisms and obesity and dyslipidemia in Thai psychotic disorder patients treated with risperidone. This study provided evidence of an association between the genetic polymorphisms of *SREBF2* (rs2267443) and *INSIG2* (rs11123469) with dyslipidemia in Thai psychotic disorder patients receiving risperidone treatment. Future functional studies are needed to establish that polymorphisms of the *SREBP2* gene and *INSIG2* contribute to the development of metabolic dysfunction. In the future, longitudinal follow-up studies with larger sample sizes are needed to confirm the roles of the *SREBF2* gene and *INSIG* polymorphisms in dyslipidemia in patients treated with risperidone.

## Figures and Tables

**Table 1 jpm-11-00943-t001:** Genotype and allele frequencies of the *SREBF2* gene and *INSIG2* (*n* = 113).

Genes	Gene Polymorphisms	Genotype	*N* (Frequency, %)	Minor Allele Frequency
*SREBF2*				
	rs1052717 (G/A)	GG	56 (49.56%)	A = 0.27
		GA	52 (46.02%)	
		AA	5 (4.42%)	
	rs2267439 (T/C)	TT	22 (19.47%)	T = 0.44
		TC	55 (48.67%)	
		CC	36 (31.86%)	
	rs2267443 (G/A)	GG	52 (46.02%)	A = 0.31
		GA	53 (46.90%)	
		AA	8 (7.08%)	
*INSIG2*				
	rs7566605 (G/C)	GG	46 (40.71%)	C = 0.40
		GC	43 (38.05%)	
		CC	24 (21.24%)	
	rs11123469 (T/C)	TT	54 (47.79%)	C = 0.30
		TC	50 (44.25%)	
		CC	9 (7.96%)	
	rs17587100 (A/C)	AA	106 (93.81%)	C = 0.03
		AC	7 (6.19%)	
		CC	0 (0.00%)	

**Table 2 jpm-11-00943-t002:** Demographic and clinical characteristics (*n* = 113).

Characteristics of the Patients	BMI < 25.0 kg/m^2^ (*n* = 56)	BMI ≥ 25.0 kg/m^2^ (*n* = 57)	*p*-Value
Age (years), median (IQR)	38.00 (31.00–47.50)	42.00 (34.00–49.50)	0.16 ^a^
Gender, *n* (%)			
Male (*n* = 53)	26 (49.06%)	27 (50.94%)	0.92 ^b^
Female (*n* = 60)	30 (50.00%)	30 (50.00%)	
Diagnosis, *n* (%)			
Schizophrenia (*n* = 92)	46 (50.00%)	46 (50.00%)	0.84 ^b^
Other diagnosis (*n* = 21)	10 (47.62%)	11 (52.38%)	
Duration of risperidone treatment (months), median (IQR)	33.75 (6.10–51.55)	22.72 (16.89–44.22)	0.78 ^a^
Dose of risperidone (mg/day), median (IQR)	4.00 (2.00–4.00)	4.00 (2.00–6.00)	0.07 ^a^
Smoking status, *n* (%)			
No	38 (48.72%)	40 (51.28%)	0.79 ^b^
Yes	18 (51.43%)	17 (48.57%)	

^a^ Data analyzed with the Mann–Whitney *U* test. ^b^ Data analyzed with the chi-square test.

**Table 3 jpm-11-00943-t003:** Effect of the *SREBF2* gene polymorphism on obesity and dyslipidemia (*n* = 113).

Obesity and Dyslipidemia	Rs1052717 (G/A)		OR (95%CI)	*p*-Value	Rs2267439 (T/C)		OR (95% CI)	*p*-Value	Rs2267443 (G/A)		OR (95% CI)	*p*-Value
	GG	GA + AA			TT	TC + CC			GG	GA + AA		
Obesity												
Absent (*n* = 56)	27 (48.21%)	29 (51.79%)	0.90 (0.43–1.88)	0.78	14 (25.00%)	42 (75.00%)	2.04 (0.78–5.34)	0.14	24 (42.86%)	32 (57.14%)	0.77 (0.37–1.63)	0.50
Present (*n* = 57)	29 (50.87%)	28 (49.13%)			8 (14.04%)	49 (85.96%)			28 (49.12%)	29 (50.88%)		
Abdominal Obesity												
Absent (*n* = 40)	21 (52.50%)	19 (47.50%)	1.20 (0.56–2.59)	0.64	8 (20.00%)	32 (80.00%)	1.05 (0.40–2.77)	0.91	20 (50.00%)	20 (50.00%)	1.28 (0.59–2.77)	0.53
Present (*n* = 73)	35 (47.95%)	38 (52.05%)			14 (19.18%)	59 (80.82%)			32 (43.84%)	41 (56.16%)		
Hypercholesterolemia												
Absent (*n* = 53)	27 (50.94%)	26 (49.06%)	1.11 (0.53–2.32)	0.78	10 (18.87%)	43 (81.13%)	0.93 (0.36–2.36)	0.88	25 (47.17%)	28 (52.83%)	1.09 (0.52–2.29)	0.82
Present (*n* =60)	29 (48.33%)	31 (51.67%)			12 (20.00%)	48 (80.00%)			27 (45.00%)	33 (55.00%)		
Hypertriglyceridemia												
Absent (*n* = 77)	42 (54.55%)	35 (45.45%)	1.89 (0.84–4.22)	0.12	18 (23.38%)	59 (76.62%)	2.44 (0.76–7.83)	0.12	41 (53.25%)	36 (46.75%)	2.56 (1.12–5.99)	0.02 *
Present (*n* = 36)	14 (38.89%)	22 (61.11%)			4 (11.11%)	32 (88.89%)			11 (30.56%)	25 (69.44%)		
Hyper-LDL Cholesterolemia												
Absent (*n* = 58)	29 (50.00%)	29 (50.00%)	1.03 (0.49–2.16)	0.92	11 (18.97%)	47 (81.03%)	0.93 (0.36–2.37)	0.89	26 (44.83%)	32 (55.17%)	0.90 (0.43-1.90)	0.79
Present (*n* = 55)	27 (49.09%)	28 (50.91%)			11 (20.00%)	44 (80.00%)			26 (47.27%)	29 (52.73%)		
Hypo-HDL Cholesterolemia												
Absent (*n* = 91)	46 (50.55%)	45 (49.45%)	1.23 (0.48–3.12)	0.67	21 (23.08%)	70 (76.92%)	6.30 (0.80–49.65)	0.05	43 (47.25%)	48 (52.75%)	1.29 (0.50–3.32)	0.59
Present (*n* = 22)	10 (45.45%)	12 (54.55%)			1 (4.55%)	21 (95.45%)			9 (40.91%)	13 (59.09%)		

Statistical significance was calculated by the chi-square test; * *p*-value < 0.05.

**Table 4 jpm-11-00943-t004:** Effect of the *INSIG2* polymorphisms on obesity and dyslipidemia (*n* = 113).

Obesity and Dyslipidemia	Rs7566605 (G/C)		OR (95% CI)	*p*-Value	Rs11123469 (T/C)		OR (95% CI)	*p*-Value	Rs17587100 (A/C)		OR (95% CI)	*p*-Value
	GG	GC + CC			TT	TC + CC			AA	AC + CC		
Obesity												
Absent (*n* = 56)	23 (41.07%)	33 (58.93%)	1.03 (0.48–2.18)	0.94	27 (48.21%)	29 (51.79%)	1.03 (0.49–2.16)	0.93	52 (92.86%)	4 (7.14%)	0.72 (0.15–3.38)	0.68
Present (*n* = 57)	23 (40.35%)	34 (59.65%)			27 (47.37%)	30 (52.63%)			54 (94.74%)	3 (5.26%)		
Abdominal Obesity												
Absent (*n* = 40)	16 (40.00%)	24 (60.00%)	0.95 (0.43–2.09)	0.91	18 (45.00%)	22 (55.00%)	0.84 (0.39–1.82)	0.66	38 (95.00%)	2 (5.00%)	1.39 (0.26–7.55)	0.69
Present (*n* = 73)	30 (41.10%)	43 (58.90%)			36 (49.32%)	37 (50.68%)			68 (93.15%)	5 (6.85%)		
Hypercholesterolemia												
Absent (*n* = 53)	24 (45.28%)	29 (54.72%)	1.42 (0.67–3.04)	0.35	26 (49.06%)	27 (50.94%)	1.10 (0.52–2.30)	0.80	49 (92.45%)	4 (7.55%)	0.64 (0.14–3.02)	0.57
Present (*n* = 60)	22 (36.67%)	38 (63.33%)			28 (46.67%)	32 (53.33%)			57 (95.00%)	3 (5.00%)		
Hypertriglyceridemia												
Absent (*n* = 77)	31 (40.26%)	46 (59.74%)	0.94 (0.42–2.10)	0.89	33 (42.86%)	44 (57.14%)	0.54 (0.27–1.06)	0.07	73 (94.81%)	4 (5.19%)	1.66 (0.35–7.83)	0.52
Present (*n* = 36)	15 (41.67%)	21 (58.33%)			21 (58.33%)	15 (41.67%)			33 (91.67%)	3 (8.33%)		
Hyper-LDL Cholesterolemia												
Absent (*n* = 58)	26 (44.83%)	32 (55.17%)	1.42 (0.67–3.02)	0.36	32 (55.17%)	26 (44.83%)	1.84 (0.87–3.89)	0.11	53 (91.38%)	5 (8.62%)	0.40 (0.07–2.15)	0.27
Present (*n* = 55)	20 (36.36%)	35 (63.64%)			22 (40.00%)	33 (60.00%)			53 (96.36%)	2 (3.64%)		
Hypo-HDL Cholesterolemia												
Absent (*n* = 91)	38 (41.76%)	53 (58.24%)	1.25 (0.48–3.29)	0.64	45 (49.45%)	46 (50.55%)	1.41 (0.55–3.63)	0.47	86 (94.51%)	5 (5.49%)	1.72 (0.31–9.51)	0.53
Present (*n* = 22)	8 (36.36%)	14 (63.64%)			9 (40.91%)	13 (59.09%)			20 (90.91%)	2 (9.09%)		

Statistical significance was calculated by the chi-square test.

**Table 5 jpm-11-00943-t005:** Multivariate logistic regression analysis of the predictive factors for risperidone-induced hypertriglyceridemia in patients with psychotic disorders (*n* = 113).

Predictive Factors	Hypertriglyceridemia		
	Odds Ratio	95% Confidence Intervals	*p*-Value
*SREBF2* rs2267439 (T/C)	2.98	0.82–10.87	0.098
*SREBF2* rs2267443 (G/A)	4.47	1.62–12.32	0.004 *
*INSIG2* rs11123469 (T/C)	0.37	0.14–0.97	0.043 *

Data are from the logistic regression analyses, a backward, conditional method. Variables entered into the method: *SREBF2* rs1052717, *SREBF2* rs2267439, *SREBF2* rs2267443, *INSIG2* rs7566605, *INSIG2* rs11123469, *INSIG2* rs17587100, age, gender, types of psychotic disorders, duration of risperidone treatment, risperidone dose, and smoking status. * *p*-value < 0.05.

## Data Availability

The datasets generated for this study will not be made publicly available, but they are available upon reasonable request to Apichaya Puangpetch (apichaya.pua@mahidol.ac.th).

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
