# Peer review of "Associations of the SREBF2 Gene and INSIG2 Polymorphisms with Obesity and Dyslipidemia in Thai Psychotic Disorder Patients Treated with Risperidone"

_jpm, 2021, doi:10.3390/jpm11100943_

Round 1

Reviewer 1 Report

The manuscript demonstrates that SNPs in SREBF2 and INSIG2 are associated with hypertriglyceridemia in patients receiving risperidone. The subject is introduced thoroughly and the data are presented clearly. 

I ask the authors a few questions to improve the manuscript. 

  1. In the section titled as "Multivariate logistic regression analysis of predictive factors for risperidone-induced hypertriglyceridemia in patients with psychotic disorders," and throughout the manuscript, authors demonstrated the association of SNPs with "risperidone-induced hypertriglyceridemia." However, I do not see any information showing that the hypertriglyceridemia in the patients are caused/induced by risperidone. I believe that the hypertriglyceridemia is rather chronic, it is unclear that the lipid pathology in these patients is caused by risperidone. If it is, I insist that the authors need to provide the data. If not, the authors cannot demonstrate that the SNPs that are researched in the manuscript are associated with risperidone-related lipid pathology. Perhaps the authors could add a section comparing the quantitative pathological risks posed by only obesity (non-risperidone-receiving psychotic patients) and by obesity/lipid disease AND risperidone-receiving conditions. 
  2. In the related question to the first one, if the study does not include risperidone-induced lipid pathology (but merely patients with lipid pathology), there are few reasons to use data from risperidone-receiving patients. To fit in the "personalized medicine" criteria, the authors should be able to show that the genetic factors posing risks to uptake risperidone-which may require a time-course study before and after receiving risperidone. I wonder if the authors could provide a statement on this question. 

Author Response

Thank you for your comments following the peer review of our report entitled " Effect of Genetic polymorphisms of SREBF2 and INSIG2 on obesity and dyslipidemia in Thai patients treated with risperidone" which was submitted to Journal of Personalized Medicine, section “Pharmacogenetics”, special issue in “Pharmacogenetics of Treating Anxiety & Depression”.

Please find the enclosed revised manuscript which has addressed the comments from the reviewer and editor. We have answered the specific reviewer's comments below in the same order that they were provided.

1. In the section titled as "Multivariate logistic regression analysis of predictive factors for risperidone-induced hypertriglyceridemia in patients with psychotic disorders," and throughout the manuscript, authors demonstrated the association of SNPs with "risperidone-induced hypertriglyceridemia." However, I do not see any information showing that the hypertriglyceridemia in the patients are caused/induced by risperidone. I believe that the hypertriglyceridemia is rather chronic, it is unclear that the lipid pathology in these patients is caused by risperidone. If it is, I insist that the authors need to provide the data. If not, the authors cannot demonstrate that the SNPs that are researched in the manuscript are associated with risperidone-related lipid pathology. Perhaps the authors could add a section comparing the quantitative pathological risks posed by only obesity (non-risperidone-receiving psychotic patients) and by obesity/lipid disease AND risperidone-receiving conditions.

Response: Thank you for the comment concerning our manuscript. Your comments are invaluable to us. I agree with your comments. Since the study is a cross-sectional study, the changes in metabolic parameters overtime related to the use of risperidone could not analyze. Also, there were no risperidone-naive psychotic patients recruited in this study. The limitation of this issue in our study was mentioned in the discussion and conclusion part. Please see on page 11, lines 122-131 in the discussion part, and on page 11, lines 142-144 in the conclusion part. However, the revised manuscript has been expanded discussion about risperidone-induced dyslipidemia. Please see on page 9, lines 25-32.

“The study of M. Murashita et al. revealed that patients on long-term risperidone monotherapy had significantly impaired body fat percentage, BMI, triglyceride, and HDL-cholesterol levels compared to healthy volunteers [38]. Consistently, a retrospective study found that risperidone-treated patients had significant weight increases, BMI, and fasting triglyceride after 1 year of treatment compared with the baseline [39]. In addition, a prospective cohort study reported that there was a significant increase in body weight and triglycerides level in the first month and continuing in the second month after patients were treated with risperidone [40].”

2. In the related question to the first one, if the study does not include risperidone-induced lipid pathology (but merely patients with lipid pathology), there are few reasons to use data from risperidone-receiving patients. To fit in the "personalized medicine" criteria, the authors should be able to show that the genetic factors posing risks to uptake risperidone-which may require a time-course study before and after receiving risperidone. I wonder if the authors could provide a statement on this question. 

Response: Thank you for your comment. In the related question and answer to the first one, the limitation of this issue in our study was mentioned in the discussion and conclusion part. Please see on page 11, lines 122-131 in the discussion part, and on page 11, lines 142-144 in the conclusion part. However, the revised manuscript has been expanded discussion about risperidone-induced dyslipidemia. Please see on page 9, lines 25-32.

Reviewer 2 Report

The authors analyzed the effect of some genetic polymorphisms of SREBF2 and INSIG2 on obesity and dyslipidemia in Thai psychotic disorder patients treated with risperidone. They presented their results in 12 pages, five tables and 37 references. They have found the A allele of the SREBF2 (rs2267443) gene a risk factor and the C allele of rs11123469 (T/C) polymorphism of INSIG2 gen with a protective effect.

I would like to add some short comments.

Page 2, line 62: instead of difference: maybe different

Page 3, line 120: Thermo Fisher Scientific should be written in parentheses.

Page 3, Statistical analysis: the criteria for adding confounding factors in the multiple logistic analysis for adjustment are not defined.

Author Response

Thank you for your comments following the peer review of our report entitled " Effect of Genetic polymorphisms of SREBF2 and INSIG2 on obesity and dyslipidemia in Thai patients treated with risperidone" which was submitted to Journal of Personalized Medicine, section “Pharmacogenetics”, special issue in “Pharmacogenetics of Treating Anxiety & Depression”.

Please find the enclosed revised manuscript which has addressed the comments from the reviewer and editor. We have answered the specific reviewer's comments below in the same order that they were provided.

1. Page 2, line 62: instead of difference: maybe different

Response: Thank you for your comment and suggestion. The word “difference” has been revised to be “maybe different” Please see on Page 2, line 62.

2. Page 3, line 120: Thermo Fisher Scientific should be written in parentheses.

Response: Thank you for your suggestion.  The sentence “DNA concentration and purification was measured with a Termo Scientifc™ NanoDrop™ spectrophotometer.” has been revised to be “DNA concentration and purification were measured with a NanoDrop spectrophotometer (Thermo Fisher Scientific, Wilmington, DE, USA).” Please see on page 3, lines 121-123.

3. Page 3, Statistical analysis: the criteria for adding confounding factors in the multiple logistic analysis for adjustment are not defined.

Response: Thank you for your comment and suggestion. The revised manuscript has been expanded the information in the statistical analysis section about confounding factors. Please see on page 3, lines 132-135.

“According to both genetic factors and non-genetic factors including age, gender, types of psychotic disorders, duration of risperidone treatment, risperidone dose, and smoking status might be risk factors for dyslipidemia, therefore, multiple logistic regression model was constructed to find factors independently associated with dyslipidemia.”

Round 2

Reviewer 1 Report

Thank you for the responses. 

Author Response

September 9, 2021

Re: Manuscript ID: jpm-1353684

Dear reviewer,

            Thank you for your comment following the peer review of our report entitled " Effect of Genetic polymorphisms of SREBF2 and INSIG2 on obesity and dyslipidemia in Thai patients treated with risperidone" which was submitted to Journal of Personalized Medicine, section “Pharmacogenetics”, special issue in “Pharmacogenetics of Treating Anxiety & Depression”.

Please find the enclosed revised manuscript which has addressed the comment from the reviewer.

  1. Please revise the title and abstract to reflect the response to reviewer 1
    that the mechanism is not necessarily associated with risperidone treatment
    since you don't know the whether the patients were obese or had dyslipidemia
    prior to starting risperidone.

Response: Thank you for the comment and suggestion. The title and abstract have been revised, since we don't know the whether the patients were obese or had dyslipidemia prior to starting risperidone. Please see on page 1.

Thank you for your consideration.

Sincerely,

Apichaya Puangpetch, Ph.D.

Division of Pharmacogenetics and Personalized Medicine, Department of Pathology,

Faculty of Medicine,Ramathibodi Hospital, Mahidol University,

Bangkok, Thailand, 10400

Tel. (66)-2-200-4331

Fax: (66)-2-200-4332
